# Cellular Aquaculture: Prospects and Challenges

**DOI:** 10.3390/mi13060828

**Published:** 2022-05-26

**Authors:** Mukunda Goswami, Yashwanth Belathur Shambhugowda, Arjunan Sathiyanarayanan, Nevil Pinto, Alexandrea Duscher, Reza Ovissipour, Wazir Singh Lakra, Ravishankar Chandragiri Nagarajarao

**Affiliations:** 1ICAR-Central Institute of Fisheries Education, Mumbai 400061, India; yashwanth.fbtpa903@cife.edu.in (Y.B.S.); sathiyanarayanan.fbtpa803@cife.edu.in (A.S.); nevilpinto1@gmail.com (N.P.); ravishankar@cife.edu.in (R.C.N.); 2Future Foods Lab and Cellular Agriculture Initiative, Department of Food Science and Technology, Seafood Agricultural Research and Extension Centre, Virginia Polytechnic Institute and State University (VT), Blacksburg, VA 24061, USA; aduscher@vt.edu (A.D.); ovissi@vt.edu (R.O.); 3NABARD Chair Unit, ICAR-Central Marine Fisheries Research Institute, Mumbai Research Centre, Mumbai 400061, India; lakraws@hotmail.com

**Keywords:** cultivated seafood, future food, aquaculture

## Abstract

Aquaculture plays an important role as one of the fastest-growing food-producing sectors in global food and nutritional security. Demand for animal protein in the form of fish has been increasing tremendously. Aquaculture faces many challenges to produce quality fish for the burgeoning world population. Cellular aquaculture can provide an alternative, climate-resilient food production system to produce quality fish. Potential applications of fish muscle cell lines in cellular aquaculture have raised the importance of developing and characterizing these cell lines. In vitro models, such as the mouse C2C12 cell line, have been extremely useful for expanding knowledge about molecular mechanisms of muscle growth and differentiation in mammals. Such studies are in an infancy stage in teleost due to the unavailability of equivalent permanent muscle cell lines, except a few fish muscle cell lines that have not yet been used for cellular aquaculture. The Prospect of cell-based aquaculture relies on the development of appropriate muscle cells, optimization of cell conditions, and mass production of cells in bioreactors. Hence, it is required to develop and characterize fish muscle cell lines along with their cryopreservation in cell line repositories and production of ideal mass cells in suitably designed bioreactors to overcome current cellular aquaculture challenges.

## 1. Introduction

A stable seafood supply is needed to maintain biodiversity to feed the world, however, it is under increasing pressure from climate change, overfishing, and pollution. Seafood, including fish, mollusks, and crustaceans, contributes to about 17% of the global demand for animal protein [1]. A substantial gap can be seen between the demand and supply of seafood in the near future with the current state of wild-capture fisheries and aquaculture [2]. The COVID-19 pandemic has increased threats to the global food system and further demonstrates the importance of resilient and sustainable animal-protein production systems [3,4]. This shows an urgent need for a new method for seafood production to build a productive, flexible, and resilient enterprise of current and future challenges. As per the report of OCED-FAO Agricultural Outlook 2020–29, aquaculture production has continued to increase by 2%, but capture fisheries production declined by 4% due to lower landings of certain major fish species, including cod, cephalopods, and some small pelagic species [5]. Global fish production is expected to exceed 200 million metric tons by 2029, increasing by 25 MMT (or 14%) from the base period (average of 2017–19), though at a slower pace (1.3% per annum) than over the previous decade (2.3% p.a.). This declined growth is driven by the effect of decreased growth rates from both capture fisheries and aquaculture [5]. By 2029, it is expected that 90% of total fish production will be utilized for direct human consumption and is projected to increase by 16.3%. However, mirroring changes in production, the rate of increase in fish available for human consumption is projected to slow from 2.5% to 1.4% p.a. in 2010–19. Growth in per capita apparent fish food consumption was reduced from 1.3% to 0.5% p.a. in 2010–19. The FAO estimated that by 2050, the world’s population will reach 9 billion; therefore, food production must increase by 70% and meat production must increase by 100% to meet global demand [6,7]. However, our current agricultural practices are not sustainable enough to address food insecurity concerns and despite efforts to address this concern, still, 1 in 8 people globally are food insecure and 1 in 6 American children may not know where their next meal is coming from. The importance of food security has been highlighted more during the current pandemic. During COVID-19, many food processors and food supply chain stakeholders were shut down, which is creating a meat shortage and increasing food insecurity concerns. In terms of sustainability, meat production is the most unsustainable food production system, which requires 40% of arable land, 36% and 29% of crops and agricultural freshwater, respectively, and is generating 14.5% of the total greenhouse gas, as well as wastes and antibiotic residues. Meat consumption must be reduced by 70% to meet the sustainable food production systems and food security requirements [8]. The consumption of plant-based proteins has been increasing rapidly. In the U.S., the number of vegans has increased by 600% from approximately 4 million in 2014 to 19.6 million in 2017. Despite emerging plant-based foods, current food production systems and agricultural practices are very far away from being sustainable. In addition, meat-eating consumers prefer meat because of its texture, flavor, and taste. Plant-based products are not compatible enough to fulfill consumers’ requirements in this regard [9]. Thus, there is an unmet need to develop novel and innovative food production systems to produce meat without raising an animal.

The world is currently moving towards a more climate-resilient food production system, such as cultured meat/in vitro meat. The interesting possibility of cultured meat was demonstrated when the cell-cultured hamburger was produced by Dr. Mark Post in 2013, and this became an active area of research globally, advancing the science of alternative proteins [10]. Potential applications of fish muscle cell lines in cellular aquaculture have raised the importance of developing and characterizing these cell lines. In vitro models, such as the C2C12 cell line, have been extremely useful for expanding knowledge about molecular mechanisms of muscle growth and differentiation in mammals. Such studies are in the infancy stage in teleost due to the unavailability of equivalent permanent muscle cell lines, except for a few fish muscle cell lines that, so far, have not been used for cellular aquaculture (Table 1). The prospect of cell-based aquaculture relies on the development of appropriate muscle cells, optimization of cell conditions, and mass productions of cells in bioreactors. Information on molecular mechanisms controlling proliferation and differentiation of precursor cells to mature myofibrils/myofibers are very much limited in fish, and hence studies on the molecular regulation of fish muscle growth would lay important groundwork for cultured fish meat production.

## 2. Scientific Challenges

### 2.1. Challenges in Seafood Sustainability

Among food commodities, seafood consumption has increased rapidly and is expected to exhibit a sharp upsurge in the next few years [24]. Seafood products play an important role in the food supply chain by providing 20% of animal-based protein and are a main source of income in many countries. With increased seafood demand, many negative environmental implications have emerged that impact the overall sustainability of oceans and food security [25]. Aquaculture and fisheries are rapidly growing industries, and their production has almost doubled in the last 10 years, with 171 million tons of production in 2016, and per capita consumption has increased to 14.9 kg, which is projected to increase to 22.5 kg by 2030 [6]. The key technical challenges in seafood sustainability are:Overfishing, illegal, unreported, and unregulated fishery activities, which are almost 20% of the total catch ($20 billion) [26].Lack of proper monitoring systems to control fraud and mislabeling. Almost 100% of red snapper is mislabeled in sushi bars, 77% in restaurants, and 88% at the retail level [27].Lack of sustainable protein sources for fish feed production, which makes aquaculture very dependent on the ocean for providing fishmeal [7].Water pollution due to intensive aquaculture activities [26].More than 40% of waste during seafood processing, which contains high-value protein, minerals, and oil, is being discarded in many cases [28].Lack of proper traceability due to the fragmented industry.Lack of accessibility to the workforce [27].Being heavily dependent on foreign suppliers; more than 90% of the seafood products in the U.S. are imported from other countries, which becomes more challenging during the pandemic with decreased control of product safety [27].

### 2.2. Challenges in Seafood Safety

Seafood is second only to vegetables in the number of sources of foodborne illnesses, and many pathogenic bacteria are naturally present in the environment, causing major issues with seafood-borne illnesses [25]. The key technical challenges [26] are:Microplastics in seafood;Harmful algal blooms (HABs) and harvesting region closure;Antibiotic-resistant bacteria and antibiotic residues.

### 2.3. Plant- and Insect-Based Protein Challenges

Several innovative technologies have been developed and applied to address food security and human health concerns, including, insect- and plant-based proteins and foods. Plant production is the most sustainable food production system, and recently, insects have gained the food industry’s attention. Insects can convert 1.5 kg of food and agricultural wastes to 1 kg biomass, making them a valuable source of protein. However, the technical gaps are:Consumer preference, and the fact that plant- and insect-based foods are not currently substitutes for conventionally-produced meat [29].A sustainable insect production system depends on food wastes, which may pose several food safety risks, which need to be addressed [30].Insects contain high amounts of chitin and lipids that need to be separated before applying as food or feed, which makes it a costly source of protein unless it could be used for producing high-value products, such as cell culture media [30,31]Plants do not contain all the required nutrients for human health and may cause health issues due to nutrient deficiency [29].

Seafood produced from fish tissue and cell cultures, referred to as cellular agriculture [32], is an emerging approach that can serve to better ensure food security in the context of climate change, increasing population, and competition for natural resources. The process of cellular agriculture was pioneered by Dr. Mark Post of Maastricht University in 2013 for the production of hamburger meat from the bovine cell [33] and was later adapted for seafood production from 2016 onwards. There are a handful of startups working on yellowtail, bluefin tuna, salmon, and shrimp products, with ongoing research focused on cell line development and refining the process of product development [34]. The production of cell-based seafood requires the extraction of muscle and fat cells from fish or shellfish, followed by their propagation in ideal conditions inside a bioreactor. Cells are typically grown on an edible scaffold that is designed to give them the structure and texture of meat [35]. The ideal result is that cultured seafood becomes indistinguishable from whole fish meat in terms of sensory experience and nutritional value of traditional seafood products [36]. Cell-based meat could help minimize the environmental issues and the effect of climate change due to current meat production. The vision of sustainable and innovative meat production reflects the cultured farm animal, a cell culture consisting of multiple cell types from the biopsy of small tissue grown on a large-scale bioreactor and then used for the production of high-quality animal protein through the induction of muscle and fat cells [35]. The application of cell-based meat production over the conventional method is shown in Figure 1.

As seafood demand drastically grows with the global population, the existing sources of seafood need to be able to support and feed the world. The activity of overfishing affects the biodiversity of oceans, and animal welfare issues impact seafood supply. The in vitro meat will provide a stable supply of seafood and also reduce the stress on the biodiversity of oceans and alleviate the consumer from the existing quality and price fluctuations. There are several cellular aquaculture companies [37] and non-profit organizations that have emerged due to technological advances and increasing concern over public health, animal welfare, and environmental issues associated with conventional aquaculture (Table 2). The historical milestone of cell-based meat production is represented in Table 3. 

## 3. Cell-Based Seafood Challenges

Cultured meat appears to be an excellent choice, as it mimics meat completely and would have many environmental, financial, and health benefits over conventional methods, and would address food security concerns. Another factor supporting cell-based seafood is from a study conducted via the Reflexive Integrative Comparative Heuristics (RICH) framework, which was applied to a variety of meat alternatives and advocated that cultured meat would require less socio-institutional and technological change than a conventional food production system [55]. This approach involves culturing cells or tissues in a synthetic environment, which substantiates its existence and promotes proliferation and growth. However, the cultivated-meat industry suffers from several unmet needs, including:Diverse available seafood and lack of available cell line sources: Despite many different available seafood species in the ocean for human consumption, only a few seafood cell lines are available, mainly for toxicology and medical studies. This limitation is due to the concept being very novel, and not that many researchers have access to seafood at different life-stages (eggs to broodstock).Lack of proper serum-free media: One of the most important factors in the formulation of cell culture media, which regulates cell growth and proliferation, is serum. Although Fetal Bovine Serum (FBS) is the most commonly used media supplement, its application for cultivated food production is limited due to the cost, possible contamination, high demand, limited supply, variation from batch to batch, inability to grow specific cells, animal suffering and environmental consequences from FBS production, high ratio of protein, and downstream processing difficulties [56].Lack of efficient techniques for optimizing the cell culture media to reduce the cost of production: Despite some efforts for optimizing media, machine learning tools have not been applied comprehensively for serum-free media optimization [57].All these limitations direct us to develop seafood cell lines from unsustainable, economically critical seafood species, optimize cost-effective serum-free media using innovative machine learning approaches by applying sustainable protein sources, and increase the yield and similarity to phenotypic characters of target seafood cells. For an effective media formulation, establishing the effects of components of media on cells and deciding its concentration accordingly would counter the negative effects of serum.

## 4. Perspectives of Cellular Aquaculture

The combination of improved cell culture and tissue engineering has facilitated the production of meat outside the body of an animal [58]. With the objective of producing meat, which is physiologically and biologically similar to conventional farm meat, these muscle tissues are produced by culturing stem cells of an organism in the controlled environment of a bioreactor with appropriate media and nutrients to differentiate itself into the muscle cell and/or fat cells and proliferate to increase the mass of meat in the scaffold [36]. The schematic representation of networking and future prospective of cellular aquaculture is represented in Figure 2. For in vitro meat production, cells, such as embryonic stem cells, and adult stem cells (i.e., microsatellite cells) are used [36,59]. The selection of stem cell type differs based on the objective, i.e., for increased regenerative capacity, satellite cells are preferred over other cells, muscle cells are preferred for the production of protein-rich meat [60], and adipose tissue-derived adult stem cells are selected for its ability to produce myogenic, osteogenic, and adipogenic cell lineages [61].

In vitro meat production has several advantages over the conventional method (animal husbandry). In terms of animal welfare and ethical concern, in vitro meat production plays a vital role in reducing slaughtering and other brutality on farming animals [62]. Buddhism and Hinduism promote a vegan diet and, in some religions, killing sacred animals is strictly prohibited. By preparing meat in the lab, to some extent, it may remove the religious barrier on meat [63]. It facilitates the manipulation of nutrients in the meat by altering the composition of culture media. Replacing omega 6 fatty acid with omega 3 fatty acid [64] and enhancing healthy fatty acid along with the addition of other essential nutrients such as vitamins increases the health benefits from the meat [65]. Conventional meat production by farming releases a significant amount of carbon and has a high requirement of energy and water; on the other hand, in vitro meat production requires less energy as it does not waste energy in physiological metabolism and unnecessary biological and social activity of living organisms [59,66,67,68], and it is known as more eco-friendly since it shows significantly reduced greenhouse gas emissions and requires less water compared to conventional meat production [69,70]. Datar and Betti described the efficient utilization of land in vitro meat production [36].

In vitro meat production is helpful in reducing foodborne diseases, such as Bovine Spongiform Encephalopathy, Swine flu, and Foot and Mouth Disease [71,72], as well as other health issues because of antibiotics and other chemicals used in conventional meat production [73,74] because of the absence of the living animal during the process of meat production. In vitro meat production reduces the time consumption in meat production. This advanced technology gives the opportunity to enhance the consumption of meat of exotic species and indirectly reduces the hunting and illegal trading of endangered animals [75].

## 5. Requirements for In Vitro Meat

### 5.1. Cell Source and Growth Factors

The most appropriate cell type for cultured meat production is the myosatellite cell. Myosatellite cells are precursors to skeletal muscle cells. They are located between the basal lamina and sarcolemma of muscle fibers. Proper identification and isolation of these cells are very important, as they are the potential cell sources for cell-based meat production. The adult stem cells are called satellite cells (SCs), which can be found in skeletal muscle tissues, and plays a role in muscle repair [76,77]. The extraction method of SCs from different species has been explained by Burton et al. [78]. The paper also summarized the comparison of SC characteristics in response to growth factors (GFs).

The process of myogenesis and proliferation of SCs can be regulated by the use of several growth factors, such as LIF [79,80], FGF [81,82], TGF-β [79,83], and insulin-like growth factor 1 (IGF-1) [79,81]. The TGF-β helps in the maturation of skeletal muscle mass and is a key regulator of intramuscular fibrogenesis [83,84]. On the other hand, the TGF-β1 suppressed the process of myogenesis in the 2D environment [85]. A biomarker for SCs, including the lineage markers, such as Pax7 and Myf5, was summarized by Yin et al. [77]. The predominant test of SCs quantified by fusion index (FI) can be estimated by assessing (1) myotube-specific staining (myogenin), (2) localization of non-specific nuclear staining (e.g., DAPI), and antibodies that stain proteins abundant in myotubes (e.g., MYH and desmin), or (3) colorimetric staining [86].

### 5.2. Scaffold

To produce 3D in vitro meat, a mechanical substratum or a scaffold is required. Scaffolds are porous biomaterials that provide key traits for linking nutrients to cells in thick tissues [87]. The scaffold must be edible and derived from non-animal sources, such as collagen, alginate, chitosan, etc., and should allow for vascularization, to develop muscle tissue. During this process, in vivo cells develop and are influenced by their interactions with the extracellular matrix (ECM). The ECM is the 3D mesh of collagen, glycoproteins, and enzymes that plays a role in transmitting biochemical and mechanical cues to the cell [88]. Scaffolds need to simulate some of the characteristics of the ECM, such as porosity, vascularization, biochemical properties, crystallinity, degradation ability, and edibility.

To promote tissue development in a 3D environment, the plurality of different cell types needs to be co-cultured to meet their natural extracellular environment. The scaffold can be composed of either a macroporous, hydrogel, or sponge-like biomaterial [89]. The macroporous biomaterial provides the mechanical support and macrostructure during tissue development, and the hydrogel helps the cells sustain in a 3D environment. Traditionally, scaffolds are designed in such a way to degrade as safe biomolecules that assimilated into the metabolic pathways, but for clean meat production, the scaffolds must be composed of biopolymers to degrade molecules with organoleptic properties achieved by chemical modification and judicious polymer selection of the scaffold [89]. The concepts behind their design, and fabrication with the proper selection of flavor-enhancing biomaterials, can be used to generate novel hydrogels and scaffolds for clean meat purposes.

The type of macroporous scaffolds may vary in their architecture, which is governed by molecular composition and fabrication techniques [90]. These can be generated from edible materials, including polysaccharides, native ECM, proteins, and decellularized plants [90,91,92]. The protein-based scaffolds of plant origin are ideal candidates for clean meat production because of their rich source of nutritional value, cytocompatibility, and low cost [93]. The architecture of a scaffold can be defined by porosity, distribution, and pore interconnectivity. The scaffold with higher porosity improves nutrient flow, cell viability, ECM deposition, cell proliferation, and cell adhesion [94].

### 5.3. Growth Medium

The growth medium for cell culture is considered a limiting factor due to an expensive price and is currently available for research purposes only and not at the recommended level for the production of in vitro meat at industrial scales [95]. The most widely used media for fish cell culture are Leibowitz Medium L-15 and Eagles Minimum Essential Medium (MEM), which are supplemented with fetal bovine serum (FBS) or fetal calf serum (FCS). FBS is the most commonly used supplement for the development of skeletal muscle with a wide concentration range from 5% to 20% [75,96,97]. The serum contains growth factors, which are helpful for cell proliferation, adhesion factors, and neutralizing trypsin activity while detaching [98]. The serum used for the cell culture also contains a wide range of hormones and growth factors [99,100]. Due to the expensive value of serum, there are some serum-free media that have been developed for specific cell types through the addition of supplementary proteins [101]. In addition to the FBS supplement, some important nutrients, such as vitamins, amino acids, trace elements, extracellular vesicles, and fatty acids, are essential for cell growth and proliferation, and must be added. There are some important growth components or growth factors used in the culture media with the recommended volume of AA2PN (vitamin c precursor), NaHCO3 (buffer), Sodium selenium, Insulin (growth factor), FGF-2 (growth factor), TGF-B (growth factor), and Transferrin (transport protein).

### 5.4. Bioreactor

Bioreactors provide an ideal environment under controlled conditions for cells by maintaining the pH, temperature, and oxygen level in the culture condition [102]. A large surface area is required to generate a sufficient number of muscle cells. In vitro meat requires large bioreactors to generate a sufficient number of muscle cells, such as stem cells and skeletal muscle. NASA has developed rotating bioreactors for the production of skeletal muscle tissue [103]. The scaffold containing cells were placed in the bioreactor for cell growth and specialization [104]. The environment of the bioreactor, such as temperature, pH, oxygen, carbon dioxide, etc., must replicate in vivo conditions. Cells can either be cultivated in fed-batch or continuous systems. The stirred tank bioreactors and fixed bed bioreactors are the most common bioreactors that are used. The stirred-tank bioreactors are used to homogenize the culture media and to facilitate the diffuser of oxygen into the culture media. This system works better for suspended cultures but can also be used for adhesive cells [105]. The fixed bed bioreactors are commonly employed for adherent cell types. They facilitate the strips of fiber that are packed together to form a bed-like surface where the cells can adhere.

### 5.5. Characterization of Cell Culture System

Misidentification of cell lines has been regarded as the major hindrance of the cell culture systems targeted for various biomedical applications. Therefore, characterization of the cell line is an integral part of cell culture systems for its species authentication and to study the integrity of the cell line. Various techniques, such as STR (Short Tandem Repeat) profiling and chromosomal and karyotyping analyses have been employed for species identification and to detect genetic abnormalities [106,107]. On the other hand, inter and intra-species contamination of cell lines affects the cultures, leading to misidentification of cell lines. Hence, utmost care must be given to identify the inter and intra-species cross-contamination by regular monitoring of cells during the proliferation period. In a study conducted out of 275 cell cultures, 36% of the cell lines were contaminated by another cell line or another species [108]. Molecular techniques, such as SNP array profiling [109], PCR amplification [110], and DNA fingerprinting [111], are some of the techniques extensively employed to detect cross-contamination.

## 6. Genetic and Molecular Markers

Genetic markers, such as species-specific antigens (immunofluorescence), isoenzyme phenotyping, chromosomal analysis, and hemagglutination (HL-A) haplotyping, have been extensively used to identify inter-species contamination. Lymphoblastoid cells are screened for the presence of the T and B cell markers [112]. No single method is capable enough to provide sufficient information to validate the characterization or authentication of any cell culture system. However, a combination of various techniques, such as species-specific antigens, isoenzyme analysis, DNA fingerprinting, and karyotyping, provides amplified information and an optimal approach to characterize the cell culture system [113]. Among the genetic marker systems, isoenzyme polymorphism analysis is considered to be the gold standard technique [110], and molecular technique multiplex STR profiling is considered a powerful method and standard method for eliminating cell lines that are misidentified [107,109].

### Gene Profiling

A computational based method of mouse cell line identification for its origin of species, cross-contamination, and aneuploidy, called “CLASP” (Cell Line Authentication by SNP Profiling) was developed [109]. It serves as a reference database to discriminate the cell lines of mouse origin commonly employed in various biotechnological applications. Similarly, STR profiling of human cell lines developed into a reference standard for human cell lines [114]. To facilitate the authentication of cell lines and also to detect cross-contamination, a novel STR similarity search tool “CLASTR” (Cell Line Authentication using STR) “The Cellosaurus” was developed [115]. DNA microsatellite STR fingerprinting of NCI-60 (National Cancer Institute- 60), a human cancer cell line panel, resulted in DNA fingerprints, which could also be a reference database [111]. Therefore, STR profiling analysis would become a necessary characterization technique for eliminating the misidentification of seafood cell lines [107]. However, alternative assays or techniques available commercially and cheaply are needed in the future for global acceptance [116]. Furthermore, a simple proteomic approach based on the Protein Expression Signatures (PES) could also be employed for the identification of fish cell lines [117]. For the first time, fish primary cultures of myocytes at different stages of proliferation and differentiation have also been characterized using a proteomic approach involving 2D gel electrophoresis and mass spectrometry [118].

## 7. Molecular-Based Studies

Cell-based studies are amongst the most important tools in cellular and molecular biology, serving as excellent model systems for studying cell physiology and biochemistry [16]. In the final product, contamination may occur due to process residuals during growth factors or scaffolding and may pose a new challenge for cellular aquaculture [119]. Cells may interact with recombinant proteins or small molecules during the early stages of production to help with growth and differentiation [118]. The selection of appropriate fish or shellfish raw materials with a lower risk of cross-contamination of pathogens; regular testing of cell banks and in-process materials to ensure free from microbes; and finally, the incorporation of steps to remove and deactivate potential undetected adventitious and endogenous viral contaminants during purification of the product [119]. A significant portion of this research, however, is misleading because the cell lines are of a different origin than the one claimed. PCR has been used to validate the cells, both in the initial primary source cells and in the cells in the cultured meat final product. In addition to PCR, product-enhanced reverse transcriptase (PERT), or immune-based assays, Short Tandem Repeat (STR) profiling, or COI gene assays, is required to test for adventitious agents (primarily pathogens) and validate cell identity [16,119]. Producing animal meat in a confined sterile environment could prevent problems including antimicrobial resistance genes, pollutants, food allergens, and zoonotic diseases, and improve food safety.

When cells begin to divide in large numbers, their genetic content is more likely to be unstable. To understand stochastic heterogeneity in cells, genomics, transcriptomics, proteomics, metabolomics, and epigenomics can be used to further characterize the cells, even for the same cell line. An in vitro assay was performed with MTT to determine the growth potential/metabolic activity of the cell line [16]. To measure the gene expression level, three complexities were used, including qRNA, microarrays (expression in fat and muscle), and RNA sequencing. Omics techniques are important to identify the traits in intramuscular fat for clean meat production [89]. Cells contain specific molecules that allow for cellular classification and sorting using fluorescence-conjugated antibodies against protein markers. Fluorescence-activated cell sorting (FACS) and magnetic-activated cell sorting (MACS) techniques could be used to select antibody-bound cells. FACS is a cell sorting and analysis method that is based on flow cytometry. Flow cytometry allows cell isolation and analysis based on physical characteristics, such as size, as well as marker expression via fluorescent signal detection [120]. In animal cells, genes were primarily focused on fatty acid composition, lipogenesis, adipogenesis, meat tenderness, and the connection between the extracellular matrix and intramuscular fat [121]. Using an RNAseq approach, upregulation of IGF2expression was observed in bovine for a long non-coding RNA molecule in the myogenesis process [122]. In farm animals, myogenesis and adipogenesis regulation was observed in naturally occurring miRNA molecules of Peroxisome proliferator-activated receptor (PPARG) and CCAAT/enhancer-binding protein (C/EBPa) expression. Such tools as microRNAs (miRNAs) and DNA methylation can be used to predict the myogenic regulating factors (MRFs) expression and activity in fish. MRFs, such as Myogenin, MyoD, and Pax7 [123], are crucial genes for controlling mechanisms of myogenin, such as cell maintenance, proliferation, and differentiation for muscle plasticity [124]. In vivo and in vitro skeletal muscle targets genes including miR-1, miR-206, and miR-133a, which are established mammalian systems [125,126]. Similarly, studies in fish have investigated, at the miRNA level, where nutrient deficiency of methionine arrested cell differentiation and expression of miR-133a was reduced [127]. After the introduction of methionine, the MRFs MyoD and Myogenin were rescued [127].

Proteome profiling of cell lines is important to identify the downstream biochemical activity of cells that is coordinated at the cellular level. Proteomics tools are useful in muscle cells for cell cycle exit, adhesion, migration, metabolism, proteolysis, ECM fusion, and muscle contraction [118]. Mass spectrometry is used to measure protein expression. In comparison to RNA measurements, downstream molecular mechanisms identify novel biomarkers, such as hypertrophy and meat tenderness [128], and establish reliable quantification of biological pathways [129], particularly for ECM deposition and posttranslational modification [130]. Proteomics research in meat focuses on skeletal muscle research and metabolic pathways associated with myogenesis and hypertrophy, as well as meat characteristics, such as color, flavor, tenderness, water holding capacity, and pH [131]. To make permanent changes in the cells of the muscle, genetic engineering was used by either introducing, removing, or rearranging DNA. In the isolation of stem cells in the muscle, established genetic markers that can help ensure cell population purity during the procedure.

## 8. Sensors, Devices, and Systems Are Available and Used in Cellular Aquaculture Ventures

The most important steps in the cultured meat process, such as step one, proliferation, are performed in cell culture dishes and flasks in monolayers. Once the number of cells is inclined, a bioreactor is used for the cells, which allows the yield of cells per unit medium volume through the highly controlled environment (temperature, dissolved oxygen, carbon dioxide, pH, and mechanical stimulation) that mimics in vivo conditions [132,133]. In general, two bioreactors are used for culturing cells, namely rocking platforms and stirred tanks [134]. Expansion bioreactors are preferred for the first phase of cells [135]. The bioreactor size and type are influenced by the passage of cells. Passaging, in the form of serial transference to reactors of inclining size, is required to satisfy the cell density [135]. In phase two of the maturation process, a tissue perfusion bioreactor was used when the 3D tissue constructs were used for production and provided uniform mixing of the media [136].

### 8.1. Bioreactor Monitor

Bioreactors were monitored and classified into three types: “offline,” “at line,” and “online.” In the offline system, a manual function is performed by eliminating a sample from a bioreactor and processing it each time in the laboratory. Even though the sample is removed from the reactor, it is tested next to it in line reactors. However, online reactors in both in situ and ex situ modes allow for testing. Furthermore, in the in situ system, the analyzer in line tests the sample (pH, dissolved oxygen, and temperature) and then returns it to the system, whereas in the ex situ strategy, the sample does not return to the bio-analyzer after being measured [137]. The most important aspects of a bioreactor are viability and cell density, which should be measured in fractions of time. To examine cells’ fluorescence or conductivity, online measurements that will be validated later with offline microscopy must be provided. In an at-line system, substrate and reagent density can be obtained from optical sensors, UV-Vis, ultrasound sensors, RAMAN spectroscopy, and fluorescence. In situ microscopy fixes to the bioreactors and acquires images of cells from the bioreactor without removing the sample. For large-scale bioprocessing, various bioreactors have been used, such as rocking motion bioreactors (500 L), stirred tank bioreactors (2000 L), hollow fiber bioreactors (1.0 × 10^9^ cells/mL), and packed bed bioreactors (5.1 × 10^8^) [138].

Recent algorithm-based analysis, i.e., computational fluid dynamics (CFD), is in high demand to analyze and solve problems that involve fluid flows. Computer systems are used to calculate the results required to simulate fluid free-stream flow and fluid interplay (liquids and gases) with surfaces defined by boundary conditions [138]. The microfluidic technique has been widely used to study biomedical applications, disease, and protein studies. Microfluidics has a wide range of applications in scale-down analysis, where convenient microfluidic platforms are integrated with sensors for the analysis of various bioprocesses to resolve issues in the scale-up process. Microfluidics recognizes the flow-induced stress on adherent cells via signal and other phenomena [139]. For long-term usage, myogenic cells are cryopreserved, which enables cell preservation for a long period in liquid nitrogen. The self-renewing capabilities of stem cells and in vitro myogenic abilities gradually decline over time. To retain these properties, cryopreservation-based cell banking is required. To keep viable cells, cryoprotectants are preferred, containing media such as dimethyl sulfoxide (DMSO), ethylene glycol, and sucrose.

### 8.2. Spoilage Detectors

The detection of meat spoilage is one of the most significant concerns in the food industry. Spoilage of meat is a metabolic process that leads to changes in sensory characteristics and is unacceptable for human consumption. The gold standard techniques, e.g., Polymerase chain reactions (PCR) and real-time PCR, allow rapid detection and identification of spoilage microbes in food products [140]. The number of spoilage indicative compounds, such as biogenic amines, carbon dioxide, or total volatile basic nitrogen, are measured, indicating spoilage and monitoring the shelf life of the meat product. For the first time, bacteria were used as a biosensor to detect meat spoilage [141]. Upon releasing volatile compounds from the spoiled meat, a downstream promoter of the engineered construct, *Bacillus subtilis,* is activated, encoding green fluorescence to indicate the spoilage. The volatile substances released are also measured by spectroscopic techniques, such as infra-red radiation, Raman spectroscopy, and hyperspectral imaging [142]. A bacterial strain, *Brochothrix thermosphactain,* from spoiled meat and seafood was detected using Matrix-Assisted Laser Desorption/Ionization/Time of Flight Mass Spectrometry (MALDI-TOF) mass spectrometry (MS). The advancement in electro-chemical-based sensors, such as e-tongue and e-nose, uses algorithms to classify food based on odor and detect pathogens.

However, the most common spoilage monitoring systems are comprised of dye-based sensors that describe the sensory characteristics of food by assessing color change that naturally occurs due to deterioration of meat [143], i.e., in simple terms, naked-eye evaluation of food to detect the spoilage. The change in food color is caused mainly due to the release of biogenic amines or TVB-N (Total volatile basic- nitrogen) generated from microorganisms causing spoilage of seafood or meat, which is an indicative component of protein-based food spoilage. A halochromic sensor was developed based on the fusion of cross-linking polymers and chlorophenol red-dodecanoic acid (CPR-DA) to monitor the change in color of the protein-based food spoilage [144].

Besides the change in food color, pH reduction is also a notable characteristic of spoiled food. Many sulfonepthaleins, such as bromophenol blue (BPB), chlorophenol red (CPR), bromocresol purple (BCP), and bromothymol blue (BTB), have been considered as the pH indicators. Recent advancements in other techniques, such as Surface-Enhanced Raman Scattering (SERS) based, aptamer-based sensors, and lab-on-chip could also be employed to assess spoilage in the meat industry [145].

## 9. Cell Line Repository

Major cell line repositories available for the scientific community at the global level are American Type Culture Collection (ATCC), European Collections of Cell Cultures (ECACC), Japanese Collection of Research Bioresources (JCRB), German Collections of Microorganism and Cell Cultures (DSMZ), Kerafast, Coriell Institute for Medical Research, and RIKEN, which catalogs and supplies cell lines globally for the scientific community emphasizing on work related to cell culture technology. “Cellosaurus” is a knowledge resource on cell lines from both vertebrates and invertebrates, providing information related to reproducibility and distribution or collections globally [146].

“CellFinder” is a single portal for accessing data related to the organ level, cellular level, and availability of mammalian cell lines based on several ontologies such as Cell Line Ontology (CLO), Cell Ontology (CO), and Experimental factor Ontology (EFO). It integrates cell line availability by a search based on literature, manual curation, and databases, such as Cellosaurus and hESCreg. It accounts for a total of 50,951 cell lines, of which 14,346 cell lines have been supplemented from literature and database searches [147]. The Department of Biotechnology (DBT), Govt. of India New Delhi has been instrumental in supporting research for the development, characterization, and conservation of fish cell lines. As the number of fish cell lines has been increasing, DBT, Govt. of India has realized the need of conserving fish cell lines in the cell line repository. DBT initially funded a project entitled Establishment of a National Repository at NBFGR, Lucknow for Conservation and Characterization of Fish Cell Lines. National Repository of Fish Cell Lines (NRFC) for maintenance and conservation of fish cell lines is in operation at NBFGR, Lucknow. GFI, USA has been supporting the development and characterization of muscle cell lines. An initiation to develop a fish muscle cell line repository will be a very encouraging step to facilitate research towards cell-based seafood meat [148].

### Importance of Cell Line Repository

The objective of any repository (Biological Resource Center) is to collect and catalog cell lines or biological specimens in one place. The major significance of the repository is the availability of characterized and quality-controlled cell lines without sparing time to develop as required at a nominal cost. The cell line repository will be instrumental for in vitro research in biotechnology and the conservation of germplasm. The repositories serve as “insurance” against the loss of cell lines within an individual laboratory. It also helps share cell resources across borders to the nations lacking resources. Cell lines of the same identity or origin from different repositories allow us to compare the data and generate research results [149].

## 10. Challenges in Cellular Aquaculture

Cell-based seafood helps to reduce the pressure on biodiversity on threatening wild species, however, many challenges remain in cell-based meat. The muscle tissue has multiple cell types with different proliferative and differentiation capabilities. The coculturing of different cell types are naturally found in muscle and can make cultured meat almost similar to natural meat. The correct identification of the proportion of such cells for co-culture is a challenging task [150]. The unavailability of crustacean and mollusk cell lines is another challenge to study the process of myogenesis. The traditional cell culture method is dependent on a serum that is a costly, inconsistent, and unsustainable component in the culture media and will be a major challenge in the large-scale production of cell-based meat [151]. The development of stem cells for meat production is the major technical challenge for in vitro meat production [152]. Meat structure is majorly dependent on the scaffold and the development of unprocessed and structured scaffold is also one of the major technical challenges of in vitro meat production [36]. The cell-surface proteins’ identification is quite challenging between muscle cells and progenitor cells [153]. A major challenge with in vitro meat production is consumer acceptance. Consumers may resist the product because of lack of knowledge about the novel technique [154] and deviation of meat color, texture, and appearance from natural meat [33]. The problem of fear about artificially made meat can be tackled by spreading awareness by scientists through open discussion and transparency about the research and advertisement by media [155]. Finally, cell-based meat should meet the desired culinary effect.

## 11. Conclusions

The future of cellular aquaculture relies on the development and characterization of appropriate muscle cell lines from commercially important, prioritized aquaculture species. Optimization of cell growth using food-grade reagents and media, facilitation of differentiation, and understanding of in vitro myogenesis are very important. Suitable bioreactors to produce mass cells that are scalable need to be designed. Integrating edible scaffolding will be important to achieve the required texture and nutrient delivery for cultivated seafood cells. Training a new generation of scientists and manufacturers for cultivated meat will be extremely important to move the field forward. Lastly, more research will need to be conducted on identifying the best methods of marketing for consumer acceptance of this safe and sustainable alternative seafood product. 

## Figures and Tables

**Figure 1 micromachines-13-00828-f001:**
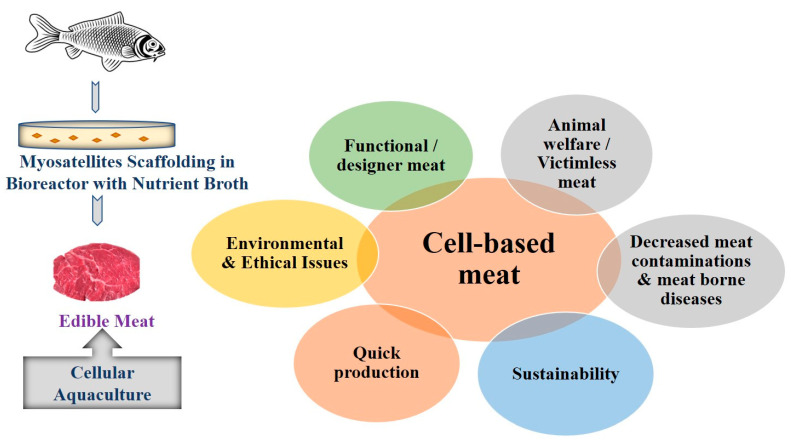
Application of cell-based meat production over conventional method.

**Figure 2 micromachines-13-00828-f002:**
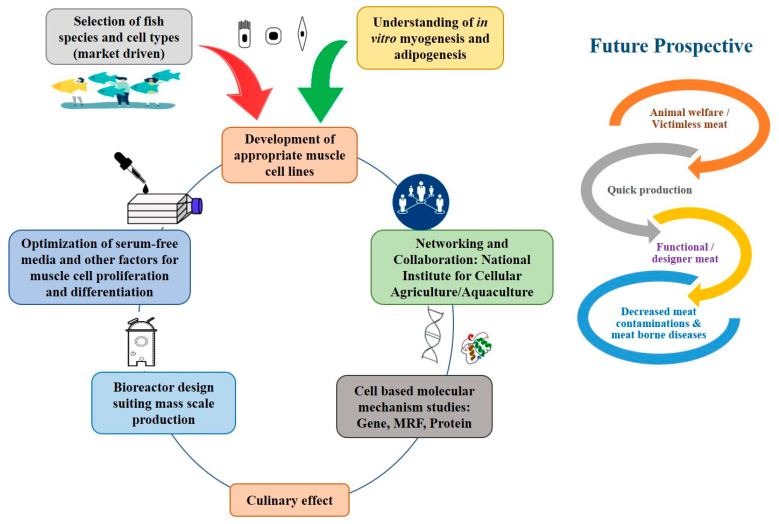
Networking and future prospective of cellular aquaculture.

**Table 1 micromachines-13-00828-t001:** Reported Cell Line Developed from Fish Muscle Tissue.

SL. No.	Cell Line	Species	Reference
1	CAM	*Carassius auratus*	[11]
2	CAM	*Cromileptes altivelis*	[12]
3	FHM	*Pimephales promelas*	[13]
4	MPCs	*Danio rerio*	[14]
5	TMF	*Scophthalmus maximus*	[15]
6	DRM	*Danio rerio*	[16]
7	MSCs	*Paralichthys olivaceus*	[17]
8	WAM	*Wallagu attu*	[18]
9	BM	*Lates calcarifer*	[19]
10	GFM	*Carassius auratus*	[20]
11	BTMS	*Caranx melampygus*	[21]
12	SHMS	*Channa striatus*	[22]
13	WSBM	*Acipenser transmontanus*	[23]

**Table 2 micromachines-13-00828-t002:** Companies working on Cellular Aquaculture.

SL. No.	Company	Headquarters	Fish Species Considered for Cultivated Seafood Production
1	Another Fish	Montreal	Whitefish
2	Avant Meats	Hong Kong	Fish maw, sea cucumber, whitefish
3	Cell Ag Tech	Ontario, Canada	Whitefish
4	Bluefin Foods	Los Angeles	Bluefin tuna
5	BlueNalu	San Diego	Tuna, mahi mahi, red snapper
6	Bluu Biosciences	Berlin	Salmon, trout, carp
7	Cultured Decadence	Madison, Wisconsin, USA	Lobster
8	Finless Foods	Emeryville, California, USA	Bluefin tuna
9	Magic Caviar	Amsterdam	Caviar
10	Memphis Meats	Berkeley, California, USA	Coho salmon
11	Sea-Stematic	Johannesburg, South Africa	–
12	Shiok Meats	Singapore	Crab, lobster, shrimp
13	SoundEats	Seattle	Whitefish, zebrafish
14	Umami Meats	Singapore	Japanese eel, red snapper, grouper, yellowfin tuna
15	Wildtype	San Francisco	Salmon

**Table 3 micromachines-13-00828-t003:** Historical milestone of in vitro meat production.

Year	Development	Reference
1912	French biologist Alexis Carrel keeps a piece of chick heart muscle alive in a Petri dish, demonstrating the possibility of keeping muscle tissue alive outside of the body.	[38]
1930	Frederick Edwin Smith, 1st Earl of Birkenhead predicts that “It will no longer be necessary to go to the extravagant length of rearing a bullock to eat its steak. From one ‘parent’ steak of choice tenderness, it will be possible to grow as large and as juicy a steak as can be desired.”	[39]
1932	Winston Churchill writes “Fifty years hence we shall escape the absurdity of growing a whole chicken to eat the breast or wing by growing these parts separately under a suitable medium.”	[39]
The early 1950s	Willem van Eelen recognizes the possibility of generating meat from tissue culture.	[38]
1971	Russell Ross achieves the in vitro cultivation of muscular fibers.	[40]
1995	The U.S. Food and Drug Administration approves the use of commercial in vitro meat production.	[41]
1999	Willem van Eelen secures the first patent for cultured meat.	[38]
2001	NASA begins in vitro meat experiments, producing cultured turkey meat.	[42]
2002	Researchers culture muscle tissue of the common goldfish in Petri dishes. The meat was judged by a test-panel to be acceptable as food.	[38]
2004	Jason Matheny founds New Harvest, the first nonprofit to work for the development of cultured meat.	[39]
2005	Dutch government agency SenterNovem begins funding cultured meat research.	[43]
2008	The In Vitro Meat Consortium holds the first international conference on the production of in vitro meat.	[44]
2008	People for the Ethical Treatment of Animals offers a $1 million prize to the first group to make a commercially viable lab-grown chicken by 2012.	[41]
2011	The company Modern Meadow, aimed at producing cultured leather and meat, is founded.	[45]
2013	The first cultured hamburger, developed by Dutch researcher Mark Post’s lab, is taste-tested.	[46]
2014	Muufri and Clara Foods, companies aimed at producing cultured dairy and eggs, respectively, are founded with the assistance of New Harvest.	[43]
2014	Real Vegan Cheese, a startup aimed at creating cultured cheese, is founded.	[47]
2014	Modern Meadow presents “steak chips”, discs of lab-grown meat that could be produced at a relatively low cost.	[45]
2015	The Modern Agriculture Foundation, which focuses on developing cultured chicken meat (as chickens make up the large majority of land animals killed for food, is founded in Israel).	[48]
2015	According to Mark Post’s lab, the cost of producing a cultured hamburger patty drops from $325,000 in 2013 to less than $12	[49]
2016	New Crop Capital, a private venture capital fund investing in alternatives to animal agriculture—including cellular agriculture—is founded. Its $25 million portfolio includes cultured meat company Memphis Meats and cultured collagen company Gelzen, along with Lighter, a software platform designed to facilitate plant-based eating, a plant-based meal delivery service called Purple Carrot, a dairy alternative called Lyrical Foods, the New Zealand plant-based meat company SunFed Foods, and alternative cheese company Miyoko’s Kitchen.	[50]
2016	The Good Food Institute, an organization devoted to promoting alternatives to animal food products—including cellular agriculture—is founded.	[51]
2016	Memphis Meats announces the creation of the first cultured meatball.	[52]
2019	Perfect Day (formerly Muufri) sells 1000 3-pint bundles of ice cream made with non-animal whey protein.	[53]
2021	Tufts University is awarded US$10 million by the USDA to establish the National Institute for Cellular Agriculture	[54]

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
