# Peer review of "Cellular Aquaculture: Prospects and Challenges"

_micromachines, 2022, doi:10.3390/mi13060828_

Round 1

Reviewer 1 Report

The review manuscript ’’Cellular Aquaculture: Prospects and Challenges’’ raises the prospect of cell-based aquaculture as an alternative system for producing quality fish. The authors have made every effort to ensure that the manuscript is appropriately organized. However, there are still opportunities to improve the manuscript. Here are some suggestions for better defining each item in this type of manuscript.

As in the chapter before (2.1. Challenges in Seafood Sustainability) and after (2.3. Plant- and Insect-Based Protein Challenges), the statements in Chapter 2.2. Challenges in Seafood Safety remain without references to the literature. It is therefore recommended that the authors also support their statements in this chapter with literature references.

In the text section lines 151 - 157 as well as in Figure 1, it is not entirely clear whether these statements are based on literature citations, as this is not evident from the failure to cite the literature.

Inappropriate font is used in several places in the text, which should be brought in line with the rest of the text, as in lines: 166 - 167, 253, 260 - 261.

Line 225: Preparing meat in the laboratory removes, to some extent, the religious barrier against meat.

Do the authors believe that eating such meat is acceptable to the religions mentioned?

The chapter 7. Molecular-based studies, in the first paragraph of this chapter, the authors have described an important part of this manuscript, but only based on two references 88 and 89. If the given text in the first paragraph is based on these two references, then it should be better structured, and if not, then this paragraph should be further supported with appropriate references.

Author Response

Author’s Response to the Reviewer’s Comments

Response to general comments

I would like to thank you for your critical evaluation of the MS and your valuable suggestions to improve the MS. You have evaluated the MS with a very strong scientific background. This has helped us to learn so many things and to improve the MS significantly. All efforts have been made to incorporate all the corrections pointed out by you and to respond to your comments.

# Reviewer 1

Response to comments

  1. As in the chapter before (1. Challenges in Seafood Sustainability) and after (2.3. Plant- and Insect-Based Protein Challenges), the statements in Chapter 2.2. Challenges in Seafood Safetyremain without references to the literature. It is therefore recommended that the authors also support their statements in this chapter with literature references.

Response: The appropriate references have been cited in Chapter 2.2 in the revised MS

  1. In the text section lines 151 - 157 as well as in Figure 1, it is not entirely clear whether these statements are based on literature citations, as this is not evident from the failure to cite the literature.

Response: The text section lines 151 – 157 was written based on the literature and cited in the revised MS and the Figure 1 was structured based on the literature and it explains the importance of cell-based meat production over the conventional method

  1. Inappropriate font is used in several places in the text, which should be brought in line with the rest of the text, as in lines: 166 - 167, 253, 260 - 261.

Response: The font size of the text, as in lines: 166 - 167, 253, 260 – 261 was thoroughly checked and corrected in the revised MS

  1. Line 225: Preparing meat in the laboratory removes, to some extent, the religious barrier against meat.

Do the authors believe that eating such meat is acceptable to the religions mentioned?

Response: In terms of animal welfare and ethical concern in vitro meat production has a vital role in reducing slaughtering and other brutality against farming animals (Weele and Driessen, 2013). Buddhism and Hinduism promote a vegan diet and, in some religions, killing sacred animals are strictly prohibited. By preparing meat in the lab to some extent it may remove the religious barrier to meat. Since it is a religious matter, not much has been written to avoid controversy.

  1. Chapter  Molecular-based studies, in the first paragraph of this chapter, the authors have described an important part of this manuscript, but only based on two references 88 and 89. If the given text in the first paragraph is based on these two references, then it should be better structured, and if not, then this paragraph should be further supported with appropriate references.

Response: Chapter 7. Molecular-based studies have been restructured with the support of references and that has been incorporated in the revised MS.

Reviewer 2 Report

This manuscript reviewed the in vitro prospects and challenges of cellular aquaculture, which is a perspective new field. The authors introduced the applications and requirement for invitro meat production. However, the relationship with the topic of this manuscript - aquaculture was not well addressed. It is suggest that the author add figures to explain bioreactor and cell culturing system working and related to these section to the cellular aquaculture.

Author Response

Author’s Response to the Reviewer’s Comments

Response to general comments

I would like to thank you for your critical evaluation of the MS and your valuable suggestions to improve the MS. You have evaluated the MS with a very strong scientific background. This has helped to improve the MS significantly. All efforts have been made to incorporate all the corrections pointed out by you and to respond to your comments.

# Reviewer 2

Comments: This manuscript reviewed the in vitro prospects and challenges of cellular aquaculture, which is a perspective new field. The authors introduced the applications and requirement for in vitro meat production. However, the relationship with the topic of this manuscript - aquaculture was not well addressed. It is suggest that the author add figures to explain bioreactor and cell culturing system working and related to these section to the cellular aquaculture.

Response: You have rightly pointed out to include the figure to explain the cell culture system working related to cellular aquaculture and we have tried our best to explain the system with graphical representation and that has been incorporated in the revised MS. The importance of cell-based meat production over the conventional method has been addressed in Figure 1 and the current trends of aquaculture in terms of production and also increase in per capita consumption have been addressed in chapter 1. As seafood demand drastically grows with the burgeoning global population, the existing sources of seafood need to be able to support and feed the world. The activity of overfishing affects the biodiversity of oceans and animal welfare issues impact the seafood supply. The in vitro meat will provide a stable supply of seafood and also reduce the stress on the biodiversity of oceans and alleviate the consumer from the existing quality and price fluctuations.
